

# Impaired emotion recognition is linked to alexithymia in heroin addicts

Giuseppe Craparo[1], Alessio Gori[2,3], Stefano Dell'Aera[4], Giulia Costanzo[1], Silvia Fasciano[1], Antonia Tomasello[1] and Carmelo M. Vicario[5,6]

[1] Faculty of Human and Social Sciences, Kore University of Enna, Italy
[2] Department of Human Sciences—Lumsa University of Rome, Italy
[3] Department of Education and Psychology—University of Florence, Italy
[4] Department of Pathological Dependences, ASP - Enna, Italy
[5] Wolfson Centre for Clinical and Cognitive Neuroscience, School of Psychology, Bangor University, Bangor, United Kingdom
[6] School of Psychology, University of Tasmania, Tasmania, Australia

## ABSTRACT

Several investigations document altered emotion processing in opiate addiction. Nevertheless, the origin of this phenomenon remains unclear. Here we examined the role of alexithymia in the ability (i.e., number of errors—*accuracy* and reaction times—*RTs*) of thirty-one heroin addicts and thirty-one healthy controls to detect several affective expressions. Results show generally lower accuracy and higher RTs in the recognition of facial expressions of emotions for patients, compared to controls. The hierarchical multivariate regression analysis shows that alexithymia might be responsible of the between groups difference with respect to the RTs in emotion detection. Overall, we provide new insights in the clinical interpretation of affective deficits in heroin addicts suggesting a role of alexithymia in their ability to recognize emotions.

Corresponding author
Carmelo M. Vicario,
carmelo.vicario@uniroma1.it,
carmelo.vicario@bangor.ac.uk

## INTRODUCTION

Drug addiction is a social-health problem with alarming statistics around the world. From a social standpoint, the addiction to drugs is frequently associated with deficient interpersonal relationships (*Kornreich et al., 2002*). This could be due to impaired processing of emotional information related to social interactions (*Kornreich et al., 2003*), though other factors such as the degree of dependence and the drug related life style may be also be involved in such impairments.

The study of emotional processing has represented a useful tool to explore interpersonal abilities in addicts, given the role of emotion processing in social attitude (e.g., *Niedenthal et al., 2005*; *Vicario & Newman, 2013*; *Vicario, Rafal & Avenanti, 2015*). Emotion recognition difficulties in opiate addicts could affect interpersonal relationships, since the ability to accurately decode facial expressions is an important component of functional/well-adjusted social interactions.

The current literature provides contrasting results about emotion recognition performance in this clinical condition (see *Kun & Demetrovics (2010)* for a review). *Kornreich et al. (2003)* studied emotion recognition in four different groups of participants: recently detoxified alcoholics (RA); opiate addicts under methadone maintenance treatment (OM); detoxified opiate addicts (OA); detoxified subjects with both alcohol and opiate dependence antecedents (DAO); Results showed a lower emotion recognition accuracy in all clinical groups, compared to healthy controls. In contrast, *Martin et al. (2006)* found that opiate users receiving methadone maintenance were more accurate than ex-opiate users in rehabilitation in recognizing facial expressions of disgust. On the other hand, they were generally slower than controls in recognizing all expressions. Finally, the study by *Zhou et al. (2012)* has shown that abstinent heroin abusers display a heightened detection of negative emotion when searching stimulus displays with a varying number of neutral faces for the positive or negative faces.

A psychological construct that could help to understand and disentangle these differences in emotional recognition performance in opiate addicts is alexithymia. The term alexithymia was coined by *Sifneos (1973)*, to indicate "*a deficit in the cognitive processing of emotions*" (see *Taylor & Bagby (2013)*). Specifically, alexithymia is characterized by a reduced ability to identify and describe feelings, a difficulty to distinguish between different feelings, an externally oriented cognitive approach to reality and a difficulty to modulate feelings (*Porcelli et al., 2004*). Alexithymia has been also associated with an impaired ability to recognize facially expressed emotions (see *Grynberg et al. (2012)* for a recent review). For example, *PrKachin, Casey & PrKachin (2009)* found an impaired ability in detecting affective expressions in populations with alexithymia. In particular, the correlational analyses documented higher difficulty in recognizing emotions such as sadness, anger, and fear. In a similar fashion, *Gil et al. (2006)* reported a significant and negative correlation between facial emotion recognition and alexithymia severity in a group of twenty patients with somatoform disorders. The research by *Lindsay & Ciarrochi (2009)* proposes two different potential explanations. One might refer to the mood, which is more negative in substance abusers (*Lindsay & Ciarrochi, 2009*), compared to controls. In fact, *Haviland et al. (1994)* found that negative mood (i.e., depression) predicts alexithymia. The alternative explanation might refer to the inaccurate belief/low-motivation of addicts. As explained by the authors, if people believe that they are not able to deal effectively with their emotions, they may be less motivated to do so. Less motivation, in turn, may lead to more alexithymic behaviour. For these reasons, one could hypothesize a key role of alexithymia in the emotional recognition deficit of addicts, given the relevance of this disorder in this clinical condition (*Craparo, 2014*; *Craparo et al., 2014a*; *Faraci et al., 2013*; *Torrado, Ouakinin & Bacelar-Nicolau, 2013*; *Craparo et al., 2014b*; *Craparo et al., 2014c*). Indeed, as reported by *Farges et al. (2014)*, the prevalence of alexithymia in addicts is 43.5%, compared to 24.6% in healthy controls, as documented by using the TAS-20. The study of alexithymia in heroin addiction is important, because it can provide insights about the origin of the emotional processing deficit in this clinical population, as reported by the literature.

The research on addicts has also provided evidence of neurobiological alterations in addicts, which might explain the emotional recognition deficit in this clinical population. For instance, *Kornreich et al. (2003)* proposed that the origin of this deficit might be due to the chronic abuse of drugs, which might cause deleterious effects on brain functions involved in decoding facial expressions. This suggestion appears likely, given the evidence of the impaired activity of several key regions for emotion (and reward) processing such as the insula, the cingulate cortex, and the amygdale in addicts (*Naqvi & Bechara, 2009*; *Di Chiara et al., 1999*; *Vicario et al., 2014*). However, these dysfunctions should not be conceived as separate from alexithymia, rather as possible neural substrates. In fact, research has linked alexithymic features to abnormal activity of the amygdale (*Kugel et al., 2008*) and the frontocingulate cortices (*Berthoz et al., 2002*).

In the current research we addressed, for the first time, the impact of alexithymia in the emotional processing deficits of heroin addicts. Indeed, despite previous studies having shown that these two phenomena are closely linked in other clinical populations such as adults with somatoform disorders (*Pedrosa Gil et al., 2009*), this remains to be investigated in heroin addicts. Thus, we measured participants' accuracy (i.e., proportion of correct answers) and reaction times (RTs) in detecting affective expressions. According to previous studies documenting a role of alexithymia in emotional recognition deficits, we expect to detect a positive relationship between alexithymia severity and the difficulty in detecting negative emotions.

## METHODS

### Participants

Sixty-two participants were selected for the current study. The drug addiction group was composed of thirty-one participants (4 cocaine/heroin addicts, 25 males, average age $34.83 \pm 8.6$); The thirty-one healthy participants (control group) was composed by 25 males, average age $33.83 \pm 8.70$). No between group difference has been detected with respect to the age ($t = 0.45$, $p = 0.65$). The four participants consuming both cocaine and heroin were excluded from the analysis in order to have a homogeneous clinical group. The clinical group was selected in two special sanitary treatment centers for drug addictions in Enna and Florence (Italy). All patients were undergoing methadone treatment. The methadone dosage ranged between 20 and 80 mg per day. No psychosocial treatment was provided. Inclusion criteria were: (i) diagnosis of heroin addiction; (ii) no previous experience of psychotherapy; (iii) no diagnosis of severe mental illness (e.g., psychosis, schizophrenia, major depression, anxiety, post-traumatic stress disorder); (iv) absence of other forms of addiction (according to the Addictive Behaviour Questionnaire, *Caretti, in press*); (v) the use of other drugs. The clinical diagnosis was made by a psychiatrist, according to the DSM V criteria, by structured clinical interview. Moreover, all patients underwent blood and urine tests to confirm the type of drug metabolites. The urine test were negative for cannabis use. The period of drug-taking ranged between 5 and 8 years. The healthy group was selected among university students and patients' relatives. They were screened through a clinical interview, aiming to mainly exclude any form of addiction.

This study was conducted in accordance with the requirements of the Helsinki convention and approved by the local ethical committee of Kore University. Informed consent was obtained from all participants.

## Measures and stimuli

The 20-item Toronto alexithymia scale (TAS-20) is a self-report scale used to measure alexithymia. It is composed of three subscales: (1) Difficulty identifying feelings (DIF); (2) Difficulty describing feelings (DDF); (3) Externally oriented thinking (EOT). *Bagby, Taylor & Parker (1994)* proposed three cut-off scores in order to discriminate alexithymic ($\geq$61), borderline (score range between 51 to 60), and non alexithymic individuals ($\leq$50). A set of pictures were used, representing the six basic emotions: happiness, sadness, fear, disgust, contempt and anger. For our sample we used an Italian validated version made by *Bressi et al. (1996)*. The scale was administered, in a multicenter research project, in a nonclinical sample of 206 adults and a nonclinical sample (medical and psychiatric disorders) of 642 subjects, showing good psychometric properties (high reliability and validity; *Bressi et al. (1996)*). In particular, as reported by the authors, the goodness-of-fit was evaluated using four criteria recommended by *Cole (1987)* and *Marsh, Balla & McDonald (1988)* viz., goodness-of-fit (GFI) $\geq$ 0.85; adjusted goodness-of-fit (AGFI) $\geq$ O.BO; the root-mean-square residual (RMSR) $\leq$ 0.10; and the Tucker-Lewis index (TLI) $\geq$ 0.80. For the normal adult sample, the (GFI) (0.88), the (AGFI) (0.84), RMSR (0.07), and TLI (0.80) all met the criteria standards, thus indicating adequacy of fit. Similar results were obtained with the clinical out-patient sample, with the GFI (0.95), AGFI (0.93) RMSR (0.05), and TLI (0.90) all meeting the criteria standards.

We used photos of easy emotional intensity level from the Facial Action Coding System (*Ekman, Friesen & Hager, 1978*). Stimuli were presented in a random order via Personal Computer.

## Procedure

Participants were invited to fill out the TAS-20 questionnaire, evaluate and categorize facial emotion expressions elicited from the photos representing the basic emotions (fear, anger, disgust, happy, sadness, surprise, contempt). They were asked to name the emotion displayed in each photo, presented in a random order and in absence of specific cues. Accuracy and reaction times (RT) (using an electronic chronometer) were recorded. The administration of both questionnaire and pictures was done in a silent room of the center for drug addicts, by using a face-to-face method. The average time of each session was about 45 min.

## Data analysis

We first used the two tailed t-test to compare the TAS-20 scores of our heroin addicts vs. control participants. Data including group and emotion as main factors were entered in a repeated-measures ANOVA to detect any between group difference with respect to the examined variables (Accuracy and RTs). Following this, we performed a hierarchical multivariate regression analysis using alexithymia (i.e., the TAS-20 score) as a predictor and emotional recognition performance (i.e., the overall average for both RTs and accuracy)

as an outcome, controlling for the age of participants and the years of exposure to heroin. Finally, we performed a Pearson correlation analysis to investigate whether the exposure to heroin predicts alexithymia severity.

For all tests, the level of statistical significance was set at $p < 0.05$. Data analyses were performed using the Statistica software, version 8.0, StatSoft, Inc., Tulsa USA and IBM SPSS Statistics 20.

## RESULTS

### Alexithymia index

As expected, we detected a significant between groups difference ($t = 4.36$, $p < 0.001$) comparing TAS-20 scores of addicts ($M = 57.6 \pm 16.7$ SD) with respect to controls ($M = 41.83 \pm 10.5$ SD). This shows that control participants are, on the average, not affected by alexithymia (i.e., TAS-20 score $\leq 50$), while addicts participants can be classified in the borderline category with respect to the alexithymia index (i.e., TAS-20 score $> 50$ and $<60$). No gender difference is reported in both groups with respect to the TAS-20 scores ($p > 0.131$). Further analyses show significant difference between the three subscales, in both the clinical ($F(2,52) = 18.52$, $p < 0.001$) and the control ($F(2,52) = 19.49$, $p < 0.001$) samples. In particular, we documented a lower score in the DIF sub-scale, compared to the DDF ($p < 0.001$) and the EOT ($p < 0.001$) scales in the clinical sample; Moreover, we documented higher scores in the EOT sub-scale, compared to the DIF ($p < 0.001$) and DDF ($p < 0.001$) subscales of the control sample.

### Emotions recognition accuracy

The repeated measure ANOVA detected a significant main effect for the Group factor ($F(1,60) = 5.68$, $p = 0.021$), documenting a lower accuracy (i.e., proportion of correct responses) for the clinical sample ($M = 0.650 \pm 0.037$) in detecting emotional stimuli compared to the control sample ($M = 0.774 \pm 0.035$). We also detected a significant main effect of the Emotion factor ($F(6,354) = 18.0$, $p < 0.001$). However, the Group $\times$ Emotion interaction term was not significant ($F(6,354) = 1.47$, $p = 0.189$). No gender difference was reported in emotion accuracy ($p = 0.851$). The Fig. 1A shows details concerning the participants' performance with respect to the accuracy in the detection of the seven emotions.

### Emotion recognition reaction times (RTs)

We detected a significant main effect for the Group factor ($F(1,55) = 4.85$, $p = 0.032$), documenting higher RTs for the clinical sample ($M = 7.74 \pm 0.608$) in detecting emotional stimuli compared to the control sample ($M = 5.90 \pm 0.567$). In a similar fashion, we documented a significant main effect for the Emotion factor ($F(6,156) = 14.7$, $p < 0.001$). In contrast, no significant difference has been reported for the Group $\times$ Emotion interaction term ($F(6,156) = 0.86$, $p = 0.522$). The Fig. 1B shows details concerning the participants' performance with respect the RTs in the detection of the seven emotions. No gender difference is reported in emotion RTs ($p = 0.639$).

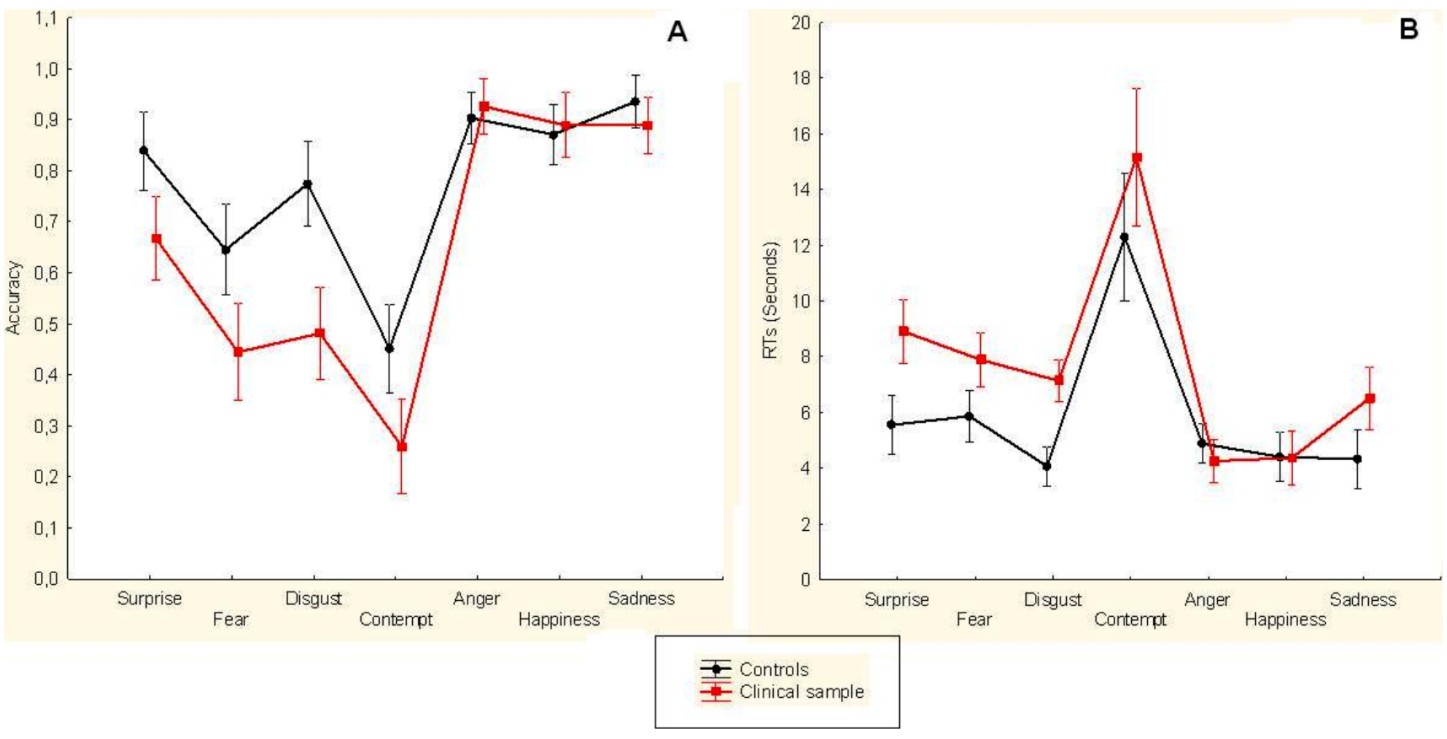

**Figure 1** (A) The figure shows the proportion of correct responses (i.e., accuracy) for healthy controls and heroin addicts (i.e., clinical sample) in the recognition of the facial expressions of emotions. The figure shows relevant differences for the recognition of Surprise, Fear, Disgust and Contempt. (B) The figure shows RTs of healthy controls and heroin addicts (i.e., clinical sample) in the recognition of the facial expressions of emotions. The figure shows relevant differences only for the detection of Surprise, Fear, Disgust, Contempt and Sadness. Vertical Bars indicate standard error.

### Hierarchical multivariate regression analysis

With respect to the RTs variable, the initial analysis (model 1, i.e., including the age and the years of exposure to heroin as predictors) provides significant results ($F(2, 57) = 4.97$, $p = 0.010$), with no effect of age on performance ($p = 0.153$), while the years of exposure to heroin was significant ($p = 0.036$). This model explain 15% of the variance. The inclusion of the alexithymia (TAS-20 score) variable as predictor (i.e., model 2, including age, years of exposure to heroin, alexithymia) cancels the effect of exposure to heroin ($p = 0.420$), while the effect of the alexithymia variable was significant ($p = 0.048$). This model explains 21% of the variance. With respect to accuracy parameters, both the model 1 ($F(2, 57) = 2.68$, $p = 0.077$) and the model 2 ($F(2, 57) = 2.40$, $p = 0.078$) were not significant.

## DISCUSSION

Several studies (*Craparo, 2014*; *Craparo et al., 2014a*; *Faraci et al., 2013*; *Torrado, Ouakinin & Bacelar-Nicolau, 2013*; *Craparo et al., 2014b*; *Craparo et al., 2014c*) have linked alexithymia to addiction. Moreover, alexithymia has been associated with deficits in emotion recognition performance (e.g., see *Taylor & Bagby, 2013*). However, the literature documents contrasting results while examining emotion recognition performance in addicts, with evidence of both lower (e.g., *Kornreich et al., 2003*) and higher (e.g., *Martin et al., 2006*; *Zhou et al., 2012*) accuracy in this population. Nevertheless, no research has

directly investigated the link between alexithymia and emotion recognition performance in heroin addicts.

Overall, in the current research we show that heroin addicts are less accurate and slower in the recognition of facial expressions of emotions, compared to healthy controls. This result corroborates the research of *Kornreich et al. (2003)* which reported a similar pattern of results, whilst appearing to contrast with the study of *Zhou et al. (2012)* who documented better performance in the recognition of negative emotions and also with the study of *Martin et al. (2006)*, who reporting higher accuracy in disgust recognition. However, our data corroborates the results of *Zhou et al. (2012)*, documenting slower RTs of heroin addicts in the overall emotion recognition, although only for the recognition of the expressions of negative emotions.

A novel result emerging from the hierarchical multivariate regression analysis, which fits with our main research hypothesis, is that alexithymia explains the between groups difference with respect to the RTs. This suggests that the poor performance of heroin addicts documented in our study can be explained, at least in part, by referring to their alexithymia traits. This is also in agreement with the research on somatoform disorder patients (*Pedrosa Gil et al., 2009*) and on Asperger syndrome (*Kätsyri et al., 2008*). Finally, we detected a positive correlation trend between years of exposure to heroin and TAS-20 scores, suggesting that the progressive use of this drug might increase alexithymia severity.

The slower performance in emotion recognition, as reported in our work, which depends on alexithymia, might originate from the effect of heroin exposure to the neural circuits that appear to be critical in emotion processing, such as the insula, the amygdala, orbitofrontal cortex, the anterior cingulated cortex and the basal ganglia (e.g., see *Adolphs (2002)*, *Vicario (2013)* and *Vicario (2016)* for a review). In fact, there is evidence (*Liu et al., 2011*; *Li et al., 2013*) of functional dysregulation of these regions in heroin abusers. Interestingly, these neural regions have been reported to be impaired in people with alexithymia (*Berthoz et al., 2002*; *Mantani et al., 2005*). Therefore, according to our correlation analysis, one could hypothesize that the long term exposure to heroin might at least exacerbate alexithymia, resulting in this effect on the neural system.

Alexithymia has been well documented across different disorders, including autism and eating disorders. This may explain, at least in part, the emotional difficulties across these populations. This leads to speculate that alexithymia intervention programs may lead to improvements in social and emotional abilities across a wide range of clinical conditions.

Future works investigating emotion processing in addiction might expand the current investigation by exploring the links between alexithymia, psychopathic traits, withdrawal symptoms and emotion recognition.

## LIMITATIONS

Several limitations should be mentioned with respect to the current study. Firstly, the distribution of alexithymia scores and the gender variable were not balanced between our two groups of participants. The clinical sample has been reduced because four participants were consuming both cocaine and heroin. Further limitations might be referred to the

absence of information about the years of scholastic education; the methadone dosage and the presence of psychopathic traits. In a similar fashion, we did not investigate psychopathological symptoms in relation to alexithymia. Finally, RTs were detected by using a stopwatch rather than a computerized system.

### Funding
The authors received no funding for this work.

### Competing Interests
The authors declare there are no competing interests.

### Author Contributions
- Giuseppe Craparo conceived and designed the experiments, performed the experiments, analyzed the data, wrote the paper, reviewed drafts of the paper.
- Alessio Gori conceived and designed the experiments.
- Stefano Dell'Area, Silvia Fasciano and Antonia Tomasello performed the experiments, contributed reagents/materials/analysis tools.
- Giulia Costanzo analyzed the data, contributed reagents/materials/analysis tools, wrote the paper.
- Carmelo M. Vicario conceived and designed the experiments, analyzed the data, wrote the paper, prepared figures and/or tables, reviewed drafts of the paper.

### Data Availability
Raw data has been uploaded as Supplemental Information.

### Supplemental Information
Supplemental information for this article can be found online at http://dx.doi.org/10.7717/peerj.1864#supplemental-information.

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
