# Peer review of "Impaired emotion recognition is linked to alexithymia in heroin addicts"

_PeerJ, doi:10.7717/peerj.1864_

## Round 0.1 · original submission · Major Revisions

· Academic Editor

Major Revisions

Thank you for your submission. The reviewers thought that the topic was of interest however major revisions are required before we can consider your paper for publication. In particular you should address the statistical issues raised, and consideration of the limitations of your work.

·

Basic reporting

no comments

Experimental design

no comments

Validity of the findings

no comments

Comments for the author

Comments on "Impaired sadness recognition is linked to alexithymia symptoms in heroin addicts" (#2015:11:7494:0:1:REVIEW) for PeerJ.

This paper makes an attempt to relate alexithymia with facial emotional decoding in heroin addiction.
However, several comments come into view:

The abstract was not properly written. There’s no need to repeat the aims of the study in the conclusion. The conclusion just duplicate the aim of the study “provide new insights…”. The authors would need to strengthen the conclusion.

It is misleading to state that study participants were recruited. How was the selection criterion?

The reliability of the results is not entirely clear. The literature showed that emotion recognition performance seems to be influenced by the clinical status of the patients (abstinent, receiving maintenance…). Therefore, this should be taken into consideration.

Data presented are limited. Please present sample characteristics since it is not clear to the reader what kind of patients are being evaluated. Are they sober? Detoxified? On maintenance program? Methadone? Buprenorphine? Naltrexone?

“Alexithymia has been also associated to impaired ability in recognize facially expressed emotions” This is the core sentence of the study and in that sense needs to be strengthen. Authors should expand more this sentence, for example, explaining possible mechanisms of involvement, in particular, in heroin addicted patients.

There is an adapted and validated version of the TAS-20? Italian version? It would be useful to indicate the psychometric characteristics of the scale in the study.

Regarding psychiatric disorders, how was the clinical diagnosis performed? DSM? Addictive Behavior Questionnaire? Please describe in detail this instrument and how it makes the diagnosis.


The sentence “The four participants using both cocaine and heroin were excluded from the analysis, to have a homogeneous clinical group” should not come in the statistical section.

Please rethink the statistical procedure. Why not evaluate the impact of alexithymia on the emotional recognition performance with a hierarchical multivariate regression analysis using alexithymia as a predictor and emotional recognition performance as an outcome, after controlling for potential confounding variables as age, gender, addiction severity or clinical status.

What is meant by alexithymia “symptoms”. I'm not sure if it is accurate to use the term “symptoms” because alexithymia is not a “formal” disorder. Please, consider changing the term throughout the paper. Also, the authors may need to modify the title of their paper.

The paper is generally clearly written but would benefit from an edit by a native English-speaking scientist or scientific editor.

In line 71, what is the significance of the statement “the incidence of alexithymia symptoms in addicts is of the 43.5%, compared to the 24.6% of healthy controls”. Incidence or prevalence? Symptoms or self-reported scores? This should be clarified.

Please verify type error in line 74.
Please verify type error in line 86.

In the introduction the author’s state: “substance addiction is frequently associated with deficient interpersonal relationships, … which could be due to impaired processing of emotional information related to social interactions”. However, being heroin addiction such a prominent social problem, it could be limited to think that these difficulties may depend (only) from emotional processing disabilities. Therefore, it would be useful to mention other variables that may influence deficient interpersonal relationships in heroin addiction. For instance, opiate dependence degree? Personality characteristics? Drug related life style?

Moreover, in line 43 the author’s state: ”The current literature provides contrasting results about the emotion recognition performance in this clinical category”. What category? Please rephrase this sentence.

In methods, the inclusion criteria were: diagnosis of heroin addiction and absence of other forms of addiction. No comorbidities with other drugs of abuse, for instance cannabis? This should be clarified.

In the paragraph of Methods section, it would be useful to indicate how the authors guarantee that the participants of the control group are “healthy”. They were screened about medical problems or drug use?

In line 205, it is not correct to suggest ”psychopathy commonly comorbid with drug addiction including heroin addiction”. Perhaps the authors mean psychopathic behavior?

It would be important to control the influence of psychopathologic symptoms in alexithymia, using for example SCL-90?

Please clarify the main goal of the study.

Please explore more why is important to study alexithymia in heroin addiction..

Considering that the control group is composed of students, there is some relationship between alexithymia and the level of education?

Is there any association with TAS-20 sub-scales?

Oddly, the study does not have limitations??

Reviewer 2 ·

Basic reporting

The manuscript is structured well. Language is professional but with numerous grammatical/linguistic errors – it should be proof-read by a native English speaker before resubmission if possible.
Background is presented clearly, and the research has a clear rationale, but key literature is missing. The Alexithymia Hypothesis, which suggests that alexithymia (rather than the disorder itself) explains emotion recognition abilities across clinical populations was first proposed by Bird and colleagues, whose work is not cited. The authors are directly testing this hypothesis, so it seems crucial that this work be referenced. Of particular relevance are the following papers:
Bird and Cook (2013) - Mixed emotions: The contribution of alexithymia to the emotional symptoms of autism. Cook et al. (2013) - Alexithymia, not autism, predicts poor recognition of emotional facial expressions. Brewer et al. (2015) - Emotion recognition deficits in eating disorders are explained by co-occurring alexithymia.
Similarly, the penultimate paragraph of the introduction should cite the finding that atypical insula function in autism may be predicted by alexithymia instead (Bird et al., 2010 - Empathic brain responses in insula are modulated by levels of alexithymia but not autism). This is likely of relevance for the atypical insula (and likely ACC and amygdala function) in heroin addicts – these atypicalities may be explained by alexithymia, rather than addiction per se, as the authors allude to.
Figures are clear, well labelled and of high quality.
The authors do not describe where raw data will be provided.

Experimental design

The question is original and within the scope of the journal. The research question is clear, and an important one. It will be clear how the research progresses our understanding of the role of alexithymia across disorder populations once the relevant literature described above is added to the paper.

The ethical standard is high. Methods are described in sufficient detail (although a figure giving an example of stimuli would be useful).

The experimental paradigm is not particularly rigorous, relying on basic accuracy and response time measures (although this remains common in the field). The design is also limited by the fact that a 7 alternative forced choice method, analysed in terms of basic accuracy and RTs. It is therefore not possible to determine whether accuracy differences reflect sensitivity or simply biased responding. Again, this is not uncommon in the literature, but should be acknowledged as a limitation.
The authors use non-parametric statistical tests, but it is not stated why this is. If the assumptions of parametric tests were not met, this should be stated (including which assumption was violated, specifically).
The clinical and control groups were not matched according to alexithymia severity, making it difficult for the authors to explicitly answer their research question (whether alexithymia accounts for emotion recognition difficulties). As the authors state, alexithymia and clinical group are entirely confounded, with the control participants exhibiting low alexithymia scores and the clinical group exhibiting high levels of alexithymia. In order to attempt to address this question, the authors should perform an ANCOVA, with the IVs Group and Emotion and the covariate Alexithymia. If alexithymia accounts for the decreased emotion recognition ability in the addiction group, one would expect the main effect of group to be reduced. Relatedly, the authors should state in the discussion that the study is limited by the fact that the groups were not explicitly matched according to alexithymia.
The correlation analyses were reported for individual emotions only. As noted above, accuracy and RT scores from a 7 alternative forced choice paradigm are subject to response bias, meaning a more reliable variable is likely to be overall accuracy and overall RT (rather than breaking these down into individual emotions).

Validity of the findings

The majority of statistical issues relate to issues of experimental design, so are described above.
The IVs (group and emotion) of the ANOVA are not explicitly stated – this should be included in the data analyses section for clarity.
The conclusions are stated clearly, and linked to the introduction. However, as described above, the authors do not seem to have explicitly tested the hypothesis they set out to. It was not possible for the authors to determine the role of alexithymia in emotion recognition across the clinical and control groups, as the two were confounded. The ANCOVA suggested above is required in order to attempt to address their hypothesis more directly, and the preferable technique of matching control and clinical groups according to alexithymia severity should be suggested for future work (see Bird et al 2010, Bird and Cook 2013, Cook et al 2013, Brewer et al 2015).
Where speculation occurs, this is explicitly stated.
Further speculation may be useful in terms of clinical implications. If the Alexithymia Hypothesis hold across disorder groups (as well as in those with autism and eating disorders), alexithymia may explain all emotional difficulties across populations, meaning a single alexithymia intervention may lead to improved social and emotional abilities across a wide range of clinical conditions.

Comments for the author

I found the hypothesis and rationale for this study very interesting, and feel that the manuscript would make an important contribution to the field. However, the statistical analyses in the current form do not lend themselves to answering the question set out (whether alexithymia can account for the deficits in emotion recognition, where observed, in those with addiction). If the authors analyse the data in the ways suggested above, they will be able to investigate their main hypothesis more explicitly.

---

## Round 0.2 · Major Revisions

· Academic Editor

Major Revisions

You have improved the earlier submission however this still requires major revisions. There are several sections where you need to provide appropriate references, and where you need to be clearer in how you describe your methodology. You should also consider the limitations of your study in response to the reviewers suggestions.

·

Basic reporting

no comments

Experimental design

adequate

Validity of the findings

no comments

Comments for the author

The authors would need to strengthen the limitation section. For instance, the samples are reduced. This should be mention.

Please reformulate this sentence in the introduction. Is confused! “substance addiction is frequently associated with deficient interpersonal relationships, especially from the interpersonal point of view”.

Please provide references for this statement: “though other factors such as the degree of dependence and the drug related life style may be also be involved in such impairments (???).

The authors state that the following features of alexithymia: ( i) a difficulty in identification and differentiation of feelings and bodily senses in a state of emotional arousal, (ii) a difficulty in describing feelings, (iii) a limited imagination and poor fantasy life, and (iv) a cognitive style focused on external reality.) are specific in addiction. Is this correct? Because these are general characteristics of alexithymia!

It is misleading to state that “blood and urine tests to confirm the type of drug addiction and exclude any further co-morbidity”. Addiction and co-morbidity can not be confirmed by blood or urine test. Only drug metabolites. Please rephrase this sentence. Moreover, the authors consider the “absence of other forms of addiction” as an inclusion criterion. But what about drug use (without addiction criteria)?? For instance, cannabis use? What were the results or urine test? All negative?

Please provide the statistical coefficients regarding the validity of TAS-20.

In the discussion, the sentence “Several studies have linked alexithymia to addiction” needs reference. If you mention STUDIES, please include the references. The same about “alexithymia has been associated with deficits in emotion recognition performance”

What were the dosages of methadone? For instance, this variable could be included in the regression analysis.

If patients are in treatment, did they receive any kind of psychosocial treatment??

Authors need to improve limitations section because this study suffers from several boundaries.

The authors mention that the clinical diagnosis was made by a psychiatrist, according the DSM V criterias, by clinical interview. What clinical interview?


The affirmation “one could hypothesize that the long term exposure to heroin might exacerbate, if not even trigger, alexithymia,” implies a great deal speculation.


The authors mention that – “We now changed the word “symptoms” with the word “status” and/or just wrote “alexithymia”. The authors have to mention if they consider alexithymia a STATE or a TRAIT!?

---

## Round 0.3 · Minor Revisions

· Academic Editor

Minor Revisions

Thank you for your swift response to the reviewers comments. I note that a couple of the references you have added to the background section are not included in your reference list. There are a few instances where your text would benefit from further proof reading. Please address these remaining minor issues.

---

## Round 0.4 · accepted · Accept

· Academic Editor

Accept

Thank you for your resubmission. This paper is now accepted for publication.